# Compressive stress gradients direct mechanoregulation of anisotropic growth in the zebrafish jaw joint

Josepha Godivier[1,2], Elizabeth A. Lawrence[3], Mengdi Wang[3], Chrissy L. Hammond[3], Niamh C. Nowlan[1,2,4]*

**1** Department of Bioengineering, Imperial College London, London, United Kingdom, **2** School of Mechanical and Materials Engineering, University College Dublin, Dublin, Ireland, **3** School of Physiology, Pharmacology & Neuroscience, University of Bristol, Bristol, United Kingdom, **4** UCD Conway Institute of Biomolecular and Biomedical Research, University College Dublin, Dublin, Ireland

* niamh.nowlan@ucd.ie

**Data Availability Statement:** All data underlying this article (confocal images, MATLAB scripts, Abaqus CAE models and simulated shapes) can be

## Abstract

Mechanical stimuli arising from fetal movements are critical factors underlying joint growth. Abnormal fetal movements negatively affect joint shape features with important implications for joint health, but the mechanisms by which mechanical forces from fetal movements influence joint growth are still unclear. In this research, we quantify zebrafish jaw joint growth in 3D in free-to-move and immobilised fish larvae between four and five days post fertilisation. We found that the main changes in size and shape in normally moving fish were in the ventrodorsal axis, while growth anisotropy was lost in the immobilised larvae. We next sought to determine the cell level activities underlying mechanoregulated growth anisotropy by tracking individual cells in the presence or absence of jaw movements, finding that the most dramatic changes in growth rates due to jaw immobility were in the ventrodorsal axis. Finally, we implemented mechanobiological simulations of joint growth with which we tested hypotheses relating specific mechanical stimuli to mechanoregulated growth anisotropy. Different types of mechanical stimulation were incorporated into the simulation to provide the mechanoregulated component of growth, in addition to the baseline (non-mechanoregulated) growth which occurs in the immobilised animals. We found that when average tissue stress over the opening and closing cycle of the joint was used as the stimulus for mechanoregulated growth, joint morphogenesis was not accurately predicted. Predictions were improved when using the stress gradients along the rudiment axes (i.e., the variation in magnitude of compression to magnitude of tension between local regions). However, the most accurate predictions were obtained when using the compressive stress gradients (i.e., the variation in compressive stress magnitude) along the rudiment axes. We conclude therefore that the dominant biophysical stimulus contributing to growth anisotropy during early joint development is the gradient of compressive stress experienced along the growth axes under cyclical loading.

accessed on Zenodo at https://doi.org/10.5281/zenodo.7586155.

**Funding:** This research was funded by an Anatomical Society PhD studentship to J.G.. C.L.H. was funded by Versus Arthritis Fellowship (Grant number 29137). E.L. was funded by a Wellcome Trust Dynamic Cell PhD studentship. M.W. was funded by the China Scholarship Council. The funders had no role in study design, data collection and analysis, decision to publish, or preparation of the manuscript.

**Competing interests:** The authors have declared that no competing interests exist.

## Author summary

The mechanical forces caused by fetal movements are important for normal development of the skeleton, and in particular for joint shape. Several common developmental musculoskeletal conditions such as developmental dysplasia of the hip and arthrogryposis are associated with reduced or restricted fetal movements. Paediatric joint malformations impair joint function and can be debilitating. To understand the origins of such conditions, it is essential to understand how the mechanical forces arising from movements influence joint growth and shape. In this research, we used a computational model of joint growth applied to the zebrafish jaw joint to study the impact of fetal movements on joint growth. We find that how the amount of compressive loading changes along the rudiment axes and over the loading cycle is critical to the normal growth of the developing joint. Our findings implicate gradients of compressive loading as a promising target when developing therapeutic strategies (such as targeted physiotherapy) for the treatment of musculoskeletal conditions.

## Introduction

Fetal movements are critical for healthy skeletal development, and abnormal movement *in utero* is associated with several paediatric conditions in which the joint does not acquire the correct shape. Developmental dysplasia of the hip and arthrogryposis are two examples of such conditions, both of which can have lasting health consequences including early onset osteoarthritis [1,2]. When skeletal muscle is absent or non-contractile in animal models, skeletal malformations include the loss of interlocking joint shape features and fusion of the skeletal elements in some (but not all) joints [3–16]. In pharmacologically paralysed chicks, for example, the femoral epiphyses are narrower both at the level of the knee [12] and of the hip [4,12] with a loss of the acetabular depth [4], while in muscleless-limb mice, the femoral condyles are smaller than those of control littermates, with abnormal protrusions [3]. Joint morphogenesis, the process by which joints acquire their shapes, is determined by co-ordinated cell activities including proliferation [5,7,8,17] and changes in cell orientation, size and intercalation [5,7,15,18]. Chondrocyte proliferation [5,7,8,18] and intercalation [5,7] are both impaired in the absence of embryonic movement. In paralysed zebrafish jaw joints and in muscleless-limb mice elbow joints, chondrocytes are generally smaller and rounder than those of controls and have an altered orientation, indicative of cell immaturity [8,15,19]. The organisation of chondrocytes into columns in the growth plate, which contributes to rudiments' elongation, is inhibited in animal models of abnormal fetal movements [7,15]. Despite observations at the tissue and cellular level, the mechanisms by which fetal movements influence joint morphogenesis are poorly understood.

Insights on cartilage mechanoregulation can be gained by studying the effects of mechanical loads on cartilage *in vivo/in ovo*, in cartilage explants *ex vivo* or chondrocytes *in vitro*. *In vivo* [5], *in ovo* [4,9] and *in vitro* [20] studies have shown that the development of functioning joints depends on the timing and duration of movement. While early movements, prior to joint cavitation (the physical separation of the skeletal elements), are crucial for the separation of joint elements [4,5,7,9,12,15], short periods of immobility after cavitation has taken place have only minor influences on joint morphology [4]. However, long periods of immobilisation result in marked shape changes which can lead to joint fusion in the most extreme cases in chick limbs [4,9,12] and larval zebrafish jaws [5,7,15]. Experiments on fetal chick knees cultured *in vitro* showed that the duration of loading is an important factor influencing joint growth and

morphogenesis, with longer durations resulting in more normally developed joints [20]. Tissue engineering research has interrogated the effects of dynamic loading on chondrocytes *in vitro*, either through direct compression or hydrostatic pressure loading. Direct compression loading promotes extracellular matrix synthesis and tissue material properties as reviewed in [21,22]. Significant increases in glycosaminoglycan (GAG) content were reported when dynamic compression was applied to juvenile bovine chondrocytes compared to unloaded controls [23,24], and cyclic hydrostatic loading has been shown to significantly increase ECM synthesis with upregulation of both GAG and collagen production [25–27]. In contrast with dynamic loading, static compression has a degenerative effect on chondrocyte metabolism leading to, for example, decreased GAG content [28–30]. While valuable insights have been gained on the specific parameters influencing chondrocyte mechanoregulation *in vitro*, the biomechanical regulation of the cells underlying joint morphogenesis remains largely unclear.

Mechanobiological simulations offer a means to integrate mechanical and biological information to bring about insights not possible with traditional approaches [31,32]. Mechanobiological models of joint growth and morphogenesis have indicated that mechanical stimuli arising from joint motion can predict the emergence of shape features seen under normal or altered loading conditions [33–37]. For example, when simulating hip joint growth, asymmetric loading conditions resulted in shape alterations of the femoral head [33–35] and the acetabulum [35] which are characteristic of shape features seen in hip dysplasia [33,35] or cerebral palsy [34]. Modelling muscle atrophy due to brachial plexus birth injury enabled the prediction of deformed glenohumeral joint shapes as seen in children [37]. A recent study of the regenerating axolotl humerus correlated interstitial pressure, driven by cyclic loading, with joint growth and shape changes [36]. However, previous mechanobiological models [33,34,36–38], including our own [35,39], have not used accurate data for cell-level inputs. The biological contribution to morphogenesis has been assumed to be proportional to chondrocyte density which was considered either uniform across the rudiment [33,34,36,37] or decreasing proportional to distance from the joint line [35,38,39]. A range of different biophysical stimuli (peak, minimum or average hydrostatic stress [33–35,38–42] and interstitial fluid pressure [36]) have been corelated with growth and morphogenesis, but a framework to quantitatively compare the relationships between specific stimuli and developmental change is lacking. The importance of measuring accurate cell-level dynamics to truly understand growth is exemplified by prior studies simulating limb bud growth and morphogenesis, in which very different theoretical growth patterns led to the same final shape [43]. To further explore the complex relationship between mechanical loading and joint morphogenesis, accurate characterisation of the contributions of cell-level dynamics to joint growth is necessary, in addition to modelling frameworks which allow the testing of hypotheses relating specific biophysical stimuli to developmental change.

Over recent years, progress has been made in characterising the cellular dynamics involved in tissue growth. Spatial morphometric analyses were conducted on light-sheet images of the embryonic murine tibia, revealing that a number of cell morphological changes and growth strategies contribute to the expansion of growth plates, and particularly spatially-dependent cell volume expansions [44]. Quantification of tissue growth based on cell lineage tracking data in the developing chick limb bud [45] and the Drosophilia wing disc [46] showed that spatially and temporally heterogeneous growth patterns coupled with growth anisotropy are major drivers of tissue morphogenesis. Recent work from our group reported that growth in the zebrafish jaw joint exhibits pronounced anisotropy likely influenced by cell orientation [47]. Integrating accurately quantified cell-level data into new mechanobiological models of joint growth will greatly deepen our understanding of the mechano-regulatory processes involved.

In this research, we aimed to identify the causal relationship between specific aspects of the biomechanical stimuli arising from embryonic movements and the patterns of joint growth. We first quantified the effects of immobility on zebrafish larval jaw joint morphogenesis from 4 to 5 days post fertilisation (dpf), finding that the normal level of growth anisotropy- in which the joint grows primarily in the ventrodorsal axis- was largely eliminated by larval immobility. We then quantified the dynamics of growth underlying normal or immobilised growth by tracking individual cells over the developmental period studied, and found that growth rates were most diminished by larval immobility along the ventrodorsal axis. Next, we implemented a mechanobiological simulation of jaw joint morphogenesis and validated that simulations using the tracked cell activities for free-to-move and immobilised larvae as the inputs for growth predicted the observed shapes correctly. Finally, we implemented simulations in which immobilised growth rates were applied to control joint shapes (serving as the baseline biological contribution to growth), supplemented by a mechanobiological component of growth in order to test hypotheses relating different types of mechanical stimuli to joint morphogenesis. Using this methodological approach, we tested which mechanical stimuli arising from jaw movements are most likely to direct mechanoregulated growth anisotropy in the zebrafish jaw joint. Based on the quantified patterns of biophysical stimuli resulting from jaw movements, we selected three distinct mechanobiological stimuli for separate incorporation into the simulation, namely; a) average stress, b) stress gradient along the growth axes (defined as how steep the variation in magnitude of compression to magnitude of tension between local regions over the loading cycle) and c) compressive stress gradient along the growth axes (the steepness of the variation in compressive stress magnitude). We find that organ-level gradients of compressive stress are likely to be a major stimulus for mechanoregulated anisotropic growth in the zebrafish jaw joint, while tension is probably not a key contributor.

## Results

### Immobilisation leads to growth rate alterations along specific anatomical axes which reflect joint shape changes

The shapes of Meckel's cartilage (MC) joint elements (shown in Fig 1A and 1B) in larvae immobilised from 3 dpf and from free-to-move larvae (controls) were measured at 4, 4.5 and 5 dpf. While there were no significant differences in MC length (anteroposterior growth), depth (ventrodorsal growth) or width (mediolateral growth) between timepoints within the free-to-move and immobilised groups, the free-to-move larvae exhibited higher increases of MC length and depth over the whole timeframe compared to immobilised larvae as seen with average shape outlines in Fig 1Cb, d. MC length increased by approximately 37% from 4 to 5 dpf in the free-to-move larvae compared to an average increase of 15% in the immobilised larvae over the same timeframe, with a significant decrease in MC length in the immobilised larvae at 5dpf (Fig 1D). Immobilised MC depth remained almost constant from 4 to 5 dpf whereas the average free-to-move MC depth increased over the same timeframe (43% increase in free-to-move larvae, compared to 6% decrease in the immobilised larvae) (Fig 1Cb, d and 1D). MC depth was significantly decreased compared to the free-to-move larvae at 5dpf (Fig 1E). The average free-to-move MC width at the level of the joint increased slightly over the investigated timeframe (11% increase) whereas the immobilised MC width remained almost unchanged over time (4% increase; Fig 1Ca, c and 1D). However, there were no significant differences between the free-to-move and immobilised groups for MC width. Therefore, growth of the depth of the MC (i.e., ventrodorsal growth) was most severely affected by the absence of jaw movements, followed by MC length, with MC width least affected.

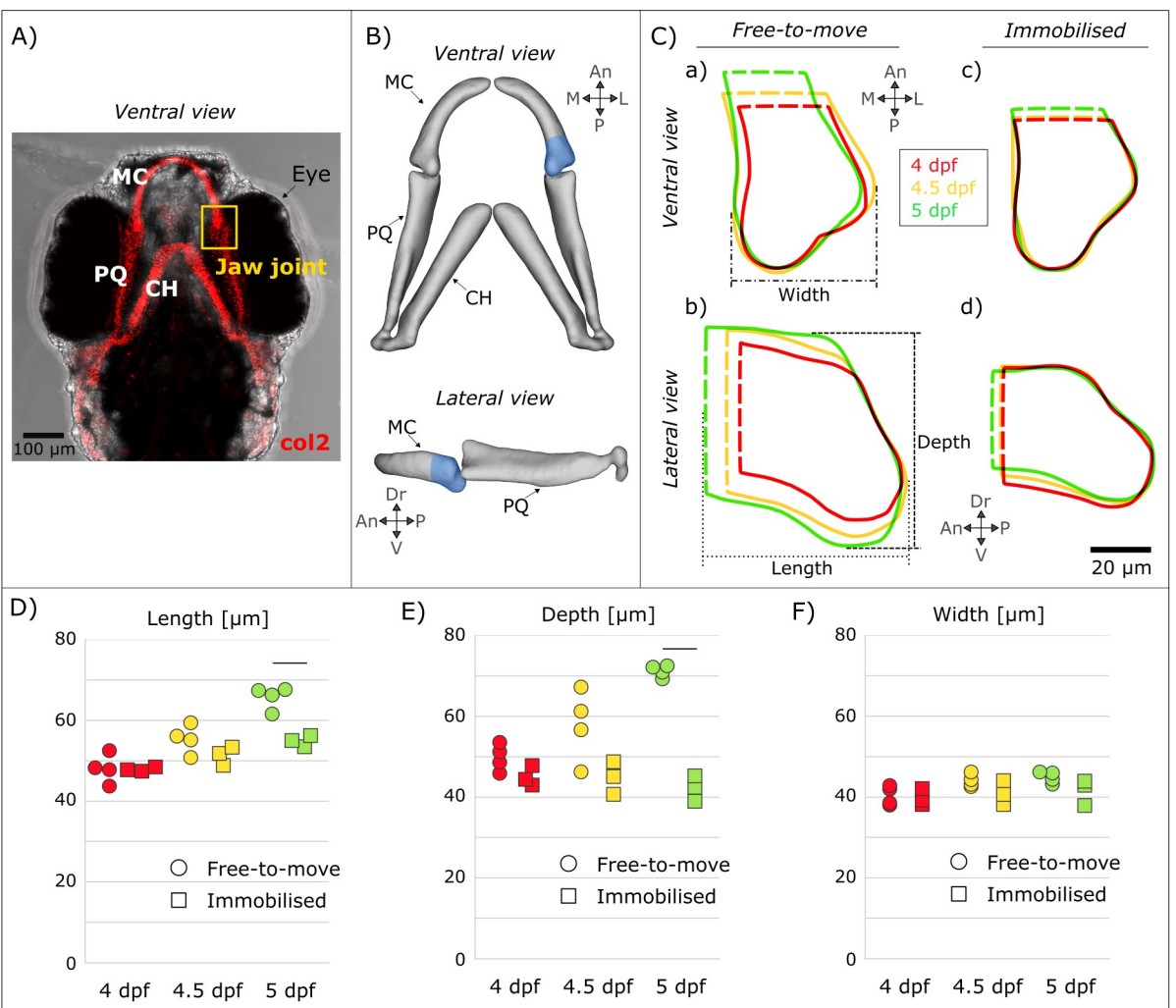

**Fig 1. In the Meckel's cartilage (MC) element, immobilisation had the most pronounced effects on rudiment depth.** (A) Brightfield ventral view of a 7 dpf zebrafish head expressing Tg(Col2a1aBAC:mCherry) cartilage marker showing the location of the jaw joint (yellow box). (B) 3D views of the jaw in the ventral and lateral planes illustrating the anterior Meckel's cartilage (MC) element. (C) Shape outlines of average MC shape at 4, 4.5 and 5 dpf for free-to-move (a, b) and immobilised (c, d) larvae. (D–F) MC length (D), depth (E) and width (F) measurements taken on individual larvae from the free-to-move (n = 4 per group) and immobilised larvae (n = 3 per group) at 4, 4.5 and 5 dpf. Bars indicate significant differences (p<0.05) between the free-to-move and immobilised groups at 5dpf; no statistically significant differences were found between the ages with each group, or between free-to-move and immobilised groups at 4 or 4.5 dpf. An: Anterior, CH: Ceratohyal, Dr: Dorsal, L: Lateral, M: Medial, MC: Meckel's cartilage, P: Posterior, PQ: Palatoquadrate, V: Ventral.

Given that different patterns of growth may result in the same shapes, without being representative of the true underlying biological processes [43], growth dynamics were quantified by tracking individual cells from 4–4.5 and from 4.5–5 dpf, following the methods described below and in [47] to characterise and visualise the actual patterns of growth which give rise to the observed organ-level shape changes. The most pronounced difference between the free-to-move and immobilised larvae was in the ventrodorsal growth rates (rudiment depth), where ventrodorsal growth rates in immobilised larvae were significantly lower in both time-windows than in free-to-move larvae as shown in Fig 2B and S1A Fig. There were no significant differences in growth rates along the anteroposterior axis (rudiment length) for either time-window (Fig 2A and S1A Fig), or for the growth along the mediolateral axis (rudiment width) from 4–4.5 dpf (Fig 2A and S1A Fig). Lastly, growth rates in the mediolateral axis were slightly,

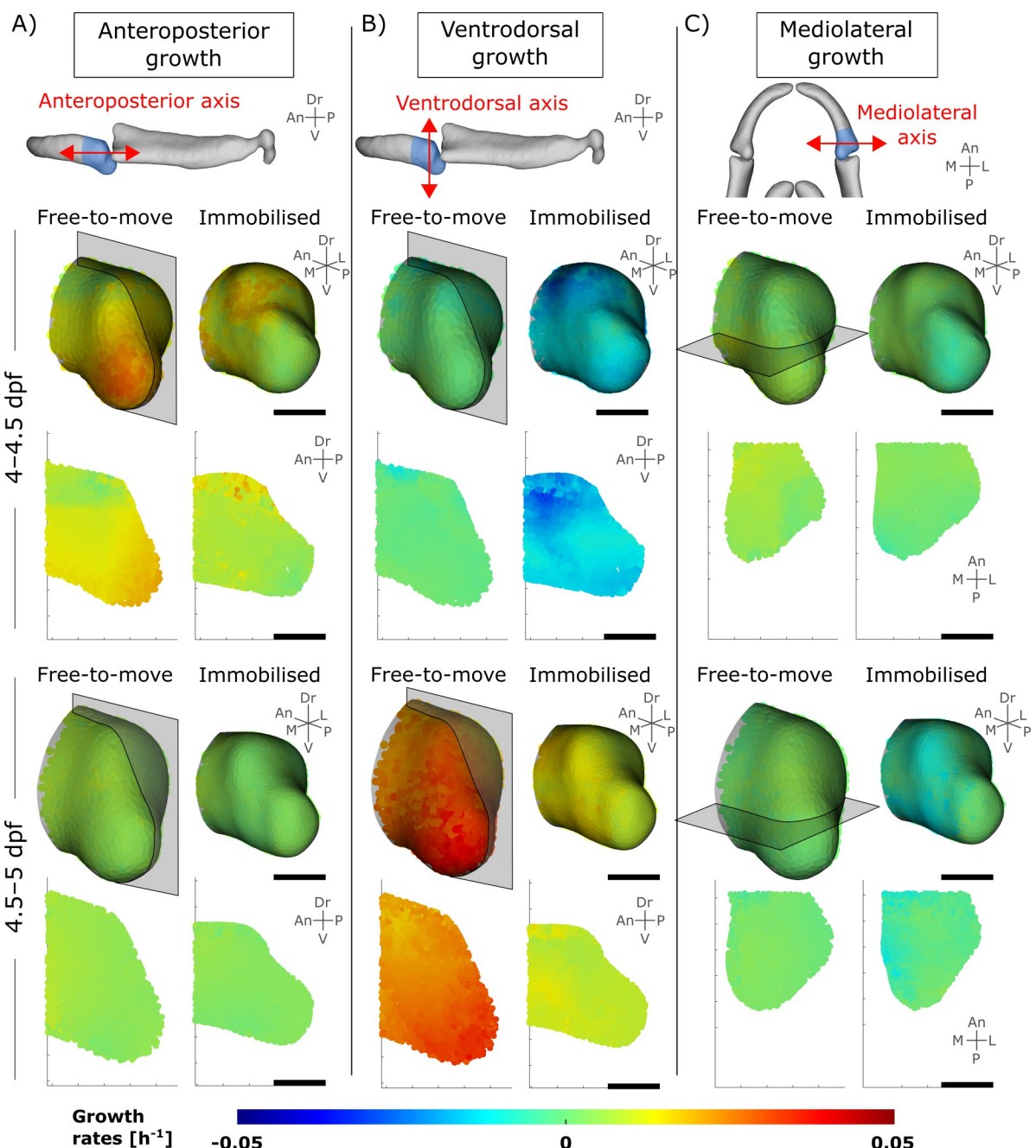

**Fig 2. Immobilisation leads to altered growth rates, primarily along the ventrodorsal axis.** Growth rates computed from tracked cells over period of interest. (A–C) Top panel: Illustration of the axes used for visualisation of growth anisotropy. Bottom panels: anteroposterior, ventrodorsal and mediolateral MC growth rates from 4–4.5 and 4.5–5 dpf for both free-to-move and immobilised larvae. Results are displayed in 3D and in one section in the lateral or ventral plane (section location shown in the 3D views). An: Anterior, Dr: Dorsal, L: Lateral, M: Medial, P: Posterior, V: Ventral.

but significantly, decreased in the immobilised larvae as compared to the free-to-move larvae from 4.5–5 dpf (Fig 2C and S1A Fig). In conclusion, the elimination of jaw movements leads to anisotropic effects on growth rates, with growth rates along the ventrodorsal axis most affected when jaw movements were absent, corresponding to the most pronounced abnormal morphological feature at the organ level.

## Computational simulations of growth

A computational simulation of MC growth was implemented using finite element (FE) methods. To validate the simulation, we first verified that simulations using the tracked cell activities for either free-to-move or immobilised larvae as the inputs for growth would predict the observed shapes correctly. Growth in free-to-move and immobilised zebrafish jaws was simulated for each 12-hour interval time window (4–4.5 and 4.5–5 dpf) following the methodology previously published [47] and described in the Methods section. The outcomes for each simulation type were consistent between the two time-windows and therefore the results for 4–4.5 dpf are provided in Fig 3, and the results from 4.5–5 dpf in S2 Fig. Growth was simulated in both scenarios using the free-to-move shape as a starting point. When growth was simulated using the measured free-to-move growth rates (as shown in Fig 3Ai and S2Ai Fig) applied to the starting shape, there was a noticeable increase in depth of the rudiment as evidenced with the expansion of the green outline as shown in Fig 3Aiii and S2Aiii Fig. However, when the measured immobilised growth rates were applied (as shown in Fig 3Aii and S2Aii Fig), no increases in depth were observed (red line in Fig 3Aiii and S2iii Fig). Therefore, our computational simulation of growth was able to replicate the most important feature found in the experimental data, which is the lack of MC depth increase in the immobilised larvae as compared to the free-to-move larvae. Furthermore, the simulation was also able to replicate the greater length increases in the free-to-move larvae compared to the immobilised larvae, with length increases more pronounced when using the free-to-move growth rates ($G_{free}$, green outlines; Figs 3Aiii and S2iii Fig) than when using the immobilised growth rates ($G_{imm}$, red outlines; Figs 3Aiii and S2iii Fig). An overlay of the simulated shape outlines from 4 dpf to 5 dpf is shown in S1B Fig.

## Changes in load patterns during jaw motion stimulate jaw joint growth anisotropy

With confidence in the implementation of our growth simulation, we next set out to test hypotheses relating different types of mechanical stimuli to joint morphogenesis. We implemented simulations in which immobilised growth rates (named $G_{imm}$) were applied to control joint shapes (serving as the baseline biological contribution to growth), supplemented by a mechanobiological component of growth. Using this methodological approach, we tested which mechanical stimuli arising from jaw movements are most likely to direct mechanoregulated growth anisotropy in the zebrafish jaw joint. We first investigated the patterns of mechanical stimuli arising from jaw motion and especially changes in hydrostatic stress patterns by performing some simple (non-growth) FE analyses. When simulating jaw opening and closing, hydrostatic stresses averaged over jaw motion were mostly spread in compression rather than in tension (Fig 4A), and a peak of compression was observed at the level of the jaw joint at both 4 and 4.5 dpf (Fig 4A, red arrowheads). At peak opening, the dorsal aspect of the MC rudiment experiences tension whereas the ventral aspect is in compression, creating a gradient in stress from tension to compression along the ventrodorsal axis (Fig 4B). At peak closure, the dorsal aspect is in compression whereas the ventral aspect is in tension, creating a reversed gradient along the ventrodorsal axis compared to peak opening (Fig 4B). Based on these data, we selected three distinct mechanobiological stimuli for separate incorporation into the simulation, namely: a) average hydrostatic stress; b) hydrostatic stress gradients (defined as how "steep" the variations in the magnitude of stresses from one aspect of the rudiment to its opposite one (done for each of the three anatomical axes)); and c) compressive hydrostatic stress gradients (defined as the stress gradients across the axes when taking into account only the compressive hydrostatic stresses).

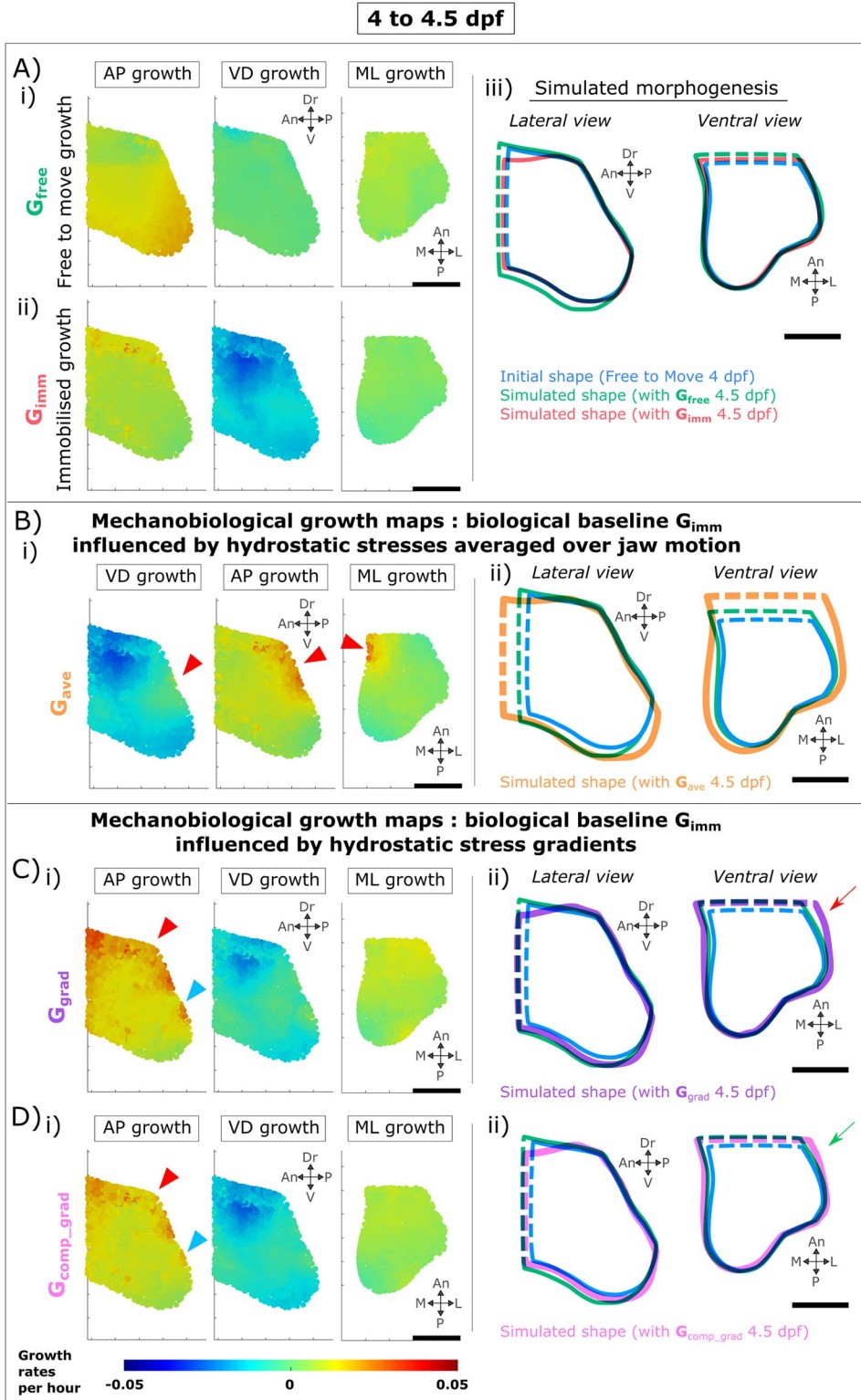

**Fig 3. Mechanobiological simulations of zebrafish larval jaw joint morphogenesis from 4–4.5 dpf incorporating different biological and mechanobiological contributions.** A) Biological contributions to morphogenesis in the absence of movements lead to undergrowth of MC depth and length compared to free-to-move shapes. A-i) Immobilised ventrodorsal (VD), anteroposterior (AP) & mediolateral (ML) growth rates applied to free-to-move 4 dpf shape. A-ii) Free-to-move growth rates. A-iii) Outlines of simulated morphogenesis with immobilised or free-to-move

growth rates promoting growth. B) Hydrostatic stresses averaged over jaw motion when used as the mechanoregulatory stimulus fail to simulate physiological jaw joint morphogenesis. B-i) Mechanobiological growth rates with compression promoting growth. Red arrowheads point to local areas of non-physiological elevated growth rates. B-ii) Outlines of simulated morphogenesis. C) Hydrostatic stress gradients as mechanobiological stimulus offer enhanced predictions of jaw joint morphogenesis. C-i) Mechanobiological growth maps in which the biological baseline $G_{imm}$ is influenced by the hydrostatic stress gradients at peak jaw opening and at peak jaw closure. Red/blue arrowheads point local areas of elevated/reduced growth rates which are not physiological. C-ii) Outlines of simulated morphogenesis. Red arrow shows MC width overgrowth. D) Compressive hydrostatic stress gradients as the stimulus for growth led to the most physiologically correct simulation of jaw joint growth. D-i) Mechanobiological growth maps in which the biological baseline $G_{imm}$ is influenced by the compressive hydrostatic stress gradients at peak opening and at peak closure. D-ii) Outlines of simulated morphogenesis. Green arrow shows the most physiological MC width as compared to previous simulations. Scale bars are 20 μm. An: Anterior, Dr: Dorsal, L: Lateral, M: Medial, P: Posterior, V: Ventral.

As above, twelve-hour time intervals were simulated: from 4 to 4.5 dpf for which results are described herein, and 4.5 to 5 dpf whose results were consistent with the first time-interval and are therefore provided in S2 Fig. In each simulation, we were seeking to compare how the shapes grown with the addition of the hypothesised mechanobiological growth rate to the immobilised (assumed "biological") growth rate would compare with the shapes grown under the normal ("free-to-move") growth rates.

We first tested the hypothesis that mechanoregulated anisotropic growth of the joint is promoted by the average hydrostatic stress over joint motion. We generated the mechanobiological growth map $G_{ave}$ as illustrated in Fig 3Bi, which exhibited spots of locally increased growth rates at the level of the joint line which were not seen in the free-to-move growth maps $G_{free}$ (red arrows in Figs 3Bi and S2Bi). From 4 to 4.5 dpf, mechanobiological simulations of joint morphogenesis using $G_{ave}$ led to a physiological increase in depth (compare the orange and green outlines in the lateral view in Fig 3Bii). However, $G_{ave}$ led to overgrowth of both MC length and MC width compared to the free-to-move simulations (Fig 3Bii). From 4.5 to 5 dpf, $G_{ave}$ led to physiological growth of MC length but was not capable of correctly predicting the physiological increases in depth or width which occur in the free-to-move larvae over that time period (S2Bii Fig). Therefore, mechanoregulated jaw joint morphogenesis could not be accurately simulated using the hydrostatic stresses averaged over joint motion as the mechanobiological stimulus.

We next investigated whether the gradient between tension and compression in the dorsal and ventral aspects of the rudiment over the opening and closing cycle (as shown in Fig 4B) could promote growth along the rudiment's depth and thus influences growth anisotropy. The spatial hydrostatic stress gradients along the three anatomical axes were calculated at peak jaw opening and closure and incorporated to $G_{imm}$ to generate a new mechanobiological growth map called $G_{grad}$ as shown in Fig 3Ci. The $G_{grad}$ growth maps from 4–4.5 and from 4.5–5 dpf were similar to the free-to-move (physiological) growth maps $G_{free}$ (Figs 3Ci and S2Ci). However, anteroposterior (AP) growth rates were higher at the dorsal aspect in $G_{grad}$ in comparison to $G_{free}$ (Figs 3Ci and S2Ci, red and blue arrowheads). From 4–4.5 dpf, $G_{grad}$ mediolateral (ML) growth rates were overall slightly higher compared to $G_{free}$ (Fig 3Ci). For both time windows, simulating morphogenesis using the mechanobiological growth map $G_{grad}$ resulted in shapes which closely resembled those obtained using the physiological free-to-move growth rates (Fig 3Cii, purple and green outlines; S2Cii Fig, purple and green outlines). However, the width of the MC distant from the joint line was bigger in the $G_{grad}$ simulated shape than the physiological shape obtained using $G_{free}$ from 4–4.5 dpf (red arrow in Fig 3Cii), while from 4.5 to 5 dpf, the shape obtained using $G_{grad}$ was slightly smaller than that of the shape obtained with $G_{free}$ (S2Cii Fig, purple and green outlines). Overall, from 4–4.5 and from 4.5–5 dpf, using the hydrostatic stress gradients arising from jaw movement as a stimulus for mechanoregulated growth, physiological jaw joint growth anisotropy was almost fully simulated.

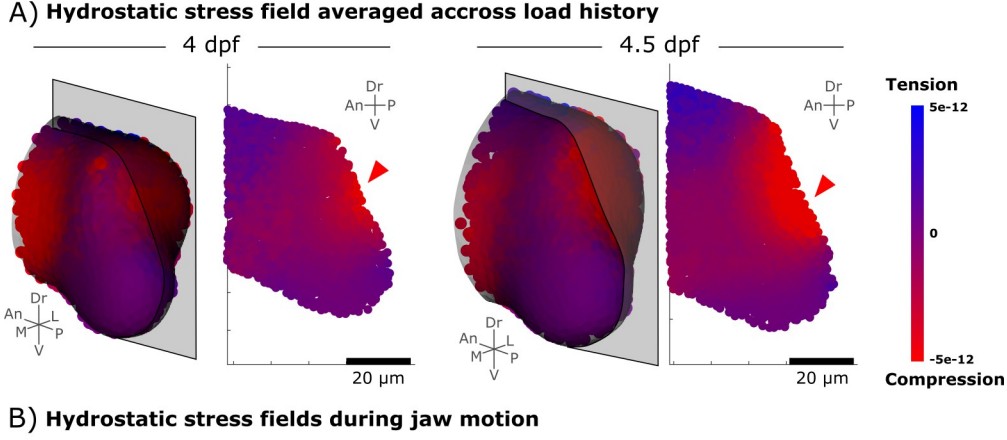

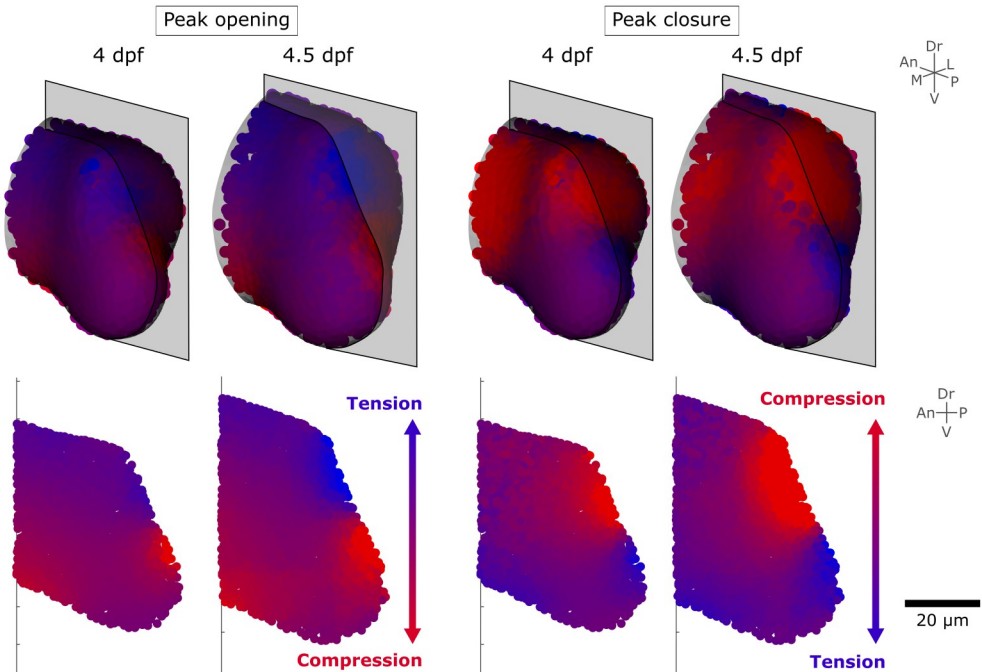

**Fig 4. Compression and tension levels arising from jaw movements.** A) Hydrostatic stress field averaged across one cycle of mouth opening and closure (termed "average stress") at 4 and 4.5 dpf. Red arrowheads indicate a peak of compression at the level of the jaw joint. Results are displayed in 3D views and in one section in the mid-lateral plane. B) Hydrostatic stress fields at peak opening and peak closure showing a change from compression to tension along the ventrodorsal axis. Results are displayed in 3D views and in one section in the mid-lateral plane. Results in the mid-ventral plane are displayed in S3 Fig. An: Anterior, Dr: Dorsal, L: Lateral, M: Medial, P: Posterior, V: Ventral.

Finally, we implemented simulations which included only the gradient to and from compression without considering the influence of hydrostatic tensile stresses. Spatial compressive hydrostatic stress gradients along the three anatomical axes at peak jaw opening and at peak jaw closure were calculated and combined with the biological baseline $G_{imm}$ to generate a new mechanobiological growth map called $G_{comp\_grad}$. The $G_{comp\_grad}$ map was more similar to the physiological growth map $G_{free}$ than the $G_{grad}$ map, with the $G_{comp\_grad}$ map having less pronounced "hot spots" of growth in the AP axis than the $G_{grad}$ map as shown with arrows in Fig 3Di. The growth simulated using $G_{comp\_grad}$ was very similar to physiological growth ($G_{free}$) and also to growth simulated using $G_{grad}$ where both tension and compression were included (Fig 3Dii), particularly from 4.5–5dpf (S2Dii Fig). However, from 4–4.5 dpf, MC

width increases were more physiological when only compressive stresses were considered ($G_{comp\_grad}$) compared to when tensile stresses were included ($G_{grad}$), (green arrow in Fig 3D-ii). Therefore, while removing the dynamic tensile components of hydrostatic stress from the mechanobiological growth algorithms did not have dramatic effects on the resultant shapes, simulations using the gradient of hydrostatic compressive stress prompted growth of shapes which most closely resembled the physiological shapes grown under normal movements. Taken together, these results indicate that organ-level gradients of compressive stress are likely to be a major stimulus for mechanoregulated anisotropic growth in the zebrafish jaw joint, while tension is probably not a key contributor.

## Discussion

In this research, growth patterns of zebrafish jaw joint morphogenesis were analysed and simulated in the presence or absence of movement. When jaw movements were absent, growth was most compromised along the ventrodorsal axis leading to pronounced decreases in MC depth in immobilised larvae compared to controls. Growth dynamics calculated from tracked cell data confirmed that growth rates were most diminished along specific anatomical axes when jaw movements were absent, verifying the anisotropic effects of mechanical loading on jaw joint morphogenesis. We constructed a mechanobiological simulation of jaw joint morphogenesis which was capable of replicating physiological growth patterns for both normal and absent movements based on tracked cell activities. We selected three potential biophysical stimuli as the main drivers for mechanoregulation of growth anisotropy, based on a finite element analysis of the stresses arising from jaw opening and closing. These three stimuli were stress averaged over jaw motion; average hydrostatic stress over joint motion, the gradient from compressive to tensile stresses over the opening and closing cycle, and the gradient to and from compressive stresses over the loading cycle (without the tensile component). Growth rates proportional to each of these stresses were combined with the baseline "biological" growth rate, which is the growth occurring in the immobilised larvae. Average stress as the mechanoregulatory component of growth did not lead to an accurate simulation of mechanoregulated jaw joint morphogenesis. In contrast, using the differences in the stresses experienced at one side of the joint compared to the opposite side (going from compression to tension) as a stimulus for mechanoregulated growth enabled almost physiological jaw joint morphogenesis to be simulated. The final simulation type, in which only the difference in compressive stresses were used as the mechanobiological stimulus, led to the most physiologically-comparable growth patterns, suggesting that the local application of compressive stresses are likely to be the main contributors to mechanoregulated growth in the zebrafish jaw joint.

We demonstrated that mechanical stimuli arising from fetal movements influence growth anisotropy in the developing joint. Chondrocyte orientation and intercalation have been shown to be affected when muscle contractions are absent in both fish [7,15] and mice [6], and we propose that the effects on organ-level growth anisotropy we report could stem from these cell level changes. In support of this theory, computational models of limb bud elongation have demonstrated that anisotropic tissue deformation strongly influences the shape of the organ during chick [45,48] and mouse [49,50] hindlimb development, and that this anisotropy is correlated with patterns in cell orientations and with a bias in the orientation of cell divisions [50].

Previous mechanobiological models of joint morphogenesis have used a range of stimuli to promote growth and shape change, including average and peak hydrostatic stress [33–35,37–40] and interstitial fluid pressure resulting from static or dynamic loading [36]. In the current research, when average hydrostatic stress distributions were used as promoters of

mechanoregulated growth, morphogenesis of the zebrafish jaw joint was not accurately predicted. Rather, we found that the dynamic changes in the patterns of mechanical stimuli along the organ axes, and especially compressive stresses, are the most likely stimuli influencing morphogenesis through altering growth anisotropy. This indicates that mechanoregulated joint growth or morphogenesis is unlikely to be determined by solely the magnitude of mechanical stimuli experienced over motion. The importance of the dynamic nature of loading concurs with *in vitro* experimental data in which static loading downregulates chondrogenesis whereas dynamic loading upregulates it [21,29,30]. The concept of a difference in biophysical stimuli within different parts of an organ (i.e., a gradient) affecting developmental change has been proposed as influencing epithelial function and pathology [51], while new bone formation in an animal model of increased loading was positively correlated with the spatial distribution of fluid pressure gradients [52]. Our combined morphological, cell-level and mechanobiological simulation analyses reveal that the growth anisotropy in the zebrafish jaw joint can only be promoted by an anisotropic growth stimulus, such as the hydrostatic gradient tested in the present work. We theorise that pronounced changes to and from high compression levels promotes fluid flow and physical deformation of the cells, leading to mechanobiological change. From our simulations of mechanoregulated growth we also conclude that tension is not a large contributor to joint growth anisotropy. The behaviour of the interzone in the un-cavitated joint as defined in our models may not be physiologically representative, and it is possible that a more accurate representation of the interzone properties would alter the distributions of compression and tension. However, our study clearly identifies the importance of anisotropic growth stimuli for anisotropic growth of the zebrafish larval jaw joint.

A strength of this research is the direct incorporation of tracked cell-level data in the mechanobiological models when previous mechanobiological simulations of joint growth used extrapolated cell data and hypothesised how they impact growth rates. Previous computational simulations, including those from our group, assumed that the biological contributions to joint growth were proportional to chondrocyte density [35,37,53]. In this research, zebrafish jaw joint growth at the macro-scale was directly quantified from tissue geometry changes which were then correlated with growth rates in each plane as determined from tracked cell-level activity. This enabled accurate quantification of the cell-level dynamics providing confidence in the underlying biological activity when testing different hypotheses regarding to how mechanical stimuli influence growth.

There are some limitations to the current work. The zebrafish jaw joint has many similarities with mammalian synovial joints [54], but cavitation occurs later in development relative to the main events of morphogenesis in other animals including mice and humans [54]. However, this research investigates a critical time of joint morphogenesis, right after movements are established, and the advantages of the zebrafish (especially the translucency of the tissues enabling live cell tracking) outweigh its disadvantages. Another limitation of the zebrafish larval model when extrapolating to mouse or human is that the jaw joint has a very small number of cells and relatively low quantity of matrix in the tissue [47], and it is possible that individual cell behaviours have a greater impact on tissue shape than in organisms with more cells and proportionally more matrix. Therefore, our conclusions could be slightly altered in larger animals, including humans. Another limitation is the use of linear elastic, rather than viscoelastic, material properties when modelling zebrafish cartilage. However, when we compared the biophysical stimuli arising from jaw movements for linear elastic vs viscoelastic properties, inconsequential differences were found, giving us confidence that the choice of material properties does not affect our findings.

In conclusion, in the absence of movement the growth anisotropy of the zebrafish jaw joint is disturbed which affects joint morphogenesis. Biophysical stimulation calculated based on

the average stress alone is not sufficient to explain the morphological changes observed at the organ-level during joint morphogenesis. Rather, changes in growth anisotropy are likely triggered by the differences in the level of compressive stresses experienced at one side of the joint compared to the opposite side, potentially leading to patterns of fluid flow which promote growth in that particular axis. Overall, this research offers avenues for improvement in simulations of joint development and potentially other organs. It provides new understanding of mechanoregulated growth in the developing joint and increases our understanding of the origins of conditions such as hip dysplasia and arthrogryposis.

## Materials and methods

### Ethics statement

Zebrafish work was approved by the Bristol AWERB (Animal Welfare and Review Board) and the UK Home Office, and was performed under Project Licence PP4700996.

### Zebrafish husbandry, lines and anaesthetisation

Fish were maintained as described previously [55,56]. All experiments were approved by the local ethics committee (Bristol Animal Welfare and Ethical Review Board) and performed under a UK Home Office Project Licence (PP4700996). Transgenic lines *Tg(col2a1aBAC: mCherry)* [57] and *Tg(-4.9sox10:eGFP)* [58] provide expression of fluorescent reporters for the immature chondrocytes in the interzone (*sox10-positive* and *col2-negative*) and the mature chondrocytes (positive for both *sox10* and *col2*). To study immobilised growth, wild type larvae were anaesthetised in 0.1 mg.ml$^{-1}$ tricaine methanesulphonate (MS222) in Danieau's buffer from 3 dpf prior to the start of recorded jaw movements [59]. The solution was refreshed twice daily until 5 dpf. At the ages studied, larval zebrafish are still dependent on the maternally deposited yolk sack for nutrients so immobilisation would not affect their ability to access nutrients. Larvae were mounted in low melting point agarose (free to move larvae were briefly immobilised for image acquisition), and imaged on a Leica sp8 confocal.

### Characterising growth from cell-level data

Growth maps were calculated for 12-hour interval time windows (4–4.5 and 4.5–5 dpf) for free-to-move and immobilised specimens following the methodology previously published [47]. In free-to-move larvae, consistent jaw movements are visible by 4 dpf [19]. Confocal image stacks centred on the jaw joint line were obtained at 4, 4.5 and 5 dpf for double transgenic *Tg(col2a1aBAC:mCherry; -4.9sox10:eGFP)* free-to-move and anaesthetised larvae as shown in Fig 5A and 5B. In immobilised specimens, cells in the posterior palatoquadrate joint element could not be reliably segmented and tracked, due to a weaker expression of fluorescent markers. Growth analyses were therefore performed solely on the anterior Meckel's cartilage (MC) joint element. Cells were segmented in Fiji [60] (Fig 5B) and the 3D cell centroid's coordinates in the MC joint element were extracted at each time point [61]. MC joint cells were manually tracked between images from two consecutive timepoints using manual labelling in MATLAB (R2018a, The MathWorks, Inc., Natick, Massachusetts, United States). Prior to the cell tracking, all rudiments were aligned in 3-matic (Materialise NV, Leuven, Belgium) at the joint line and retroarticular process (a distinctive shape feature at the ventral side of the MC [62]) and transformation matrices describing this alignment were exported. Transformation matrices exported from 3-matic were applied accordingly to align the cell centroids of each dataset in MATLAB. Approximately 30 cells constitute the portion of the rudiment modelled, with minimal proliferation over the 24 hour period studied [47].

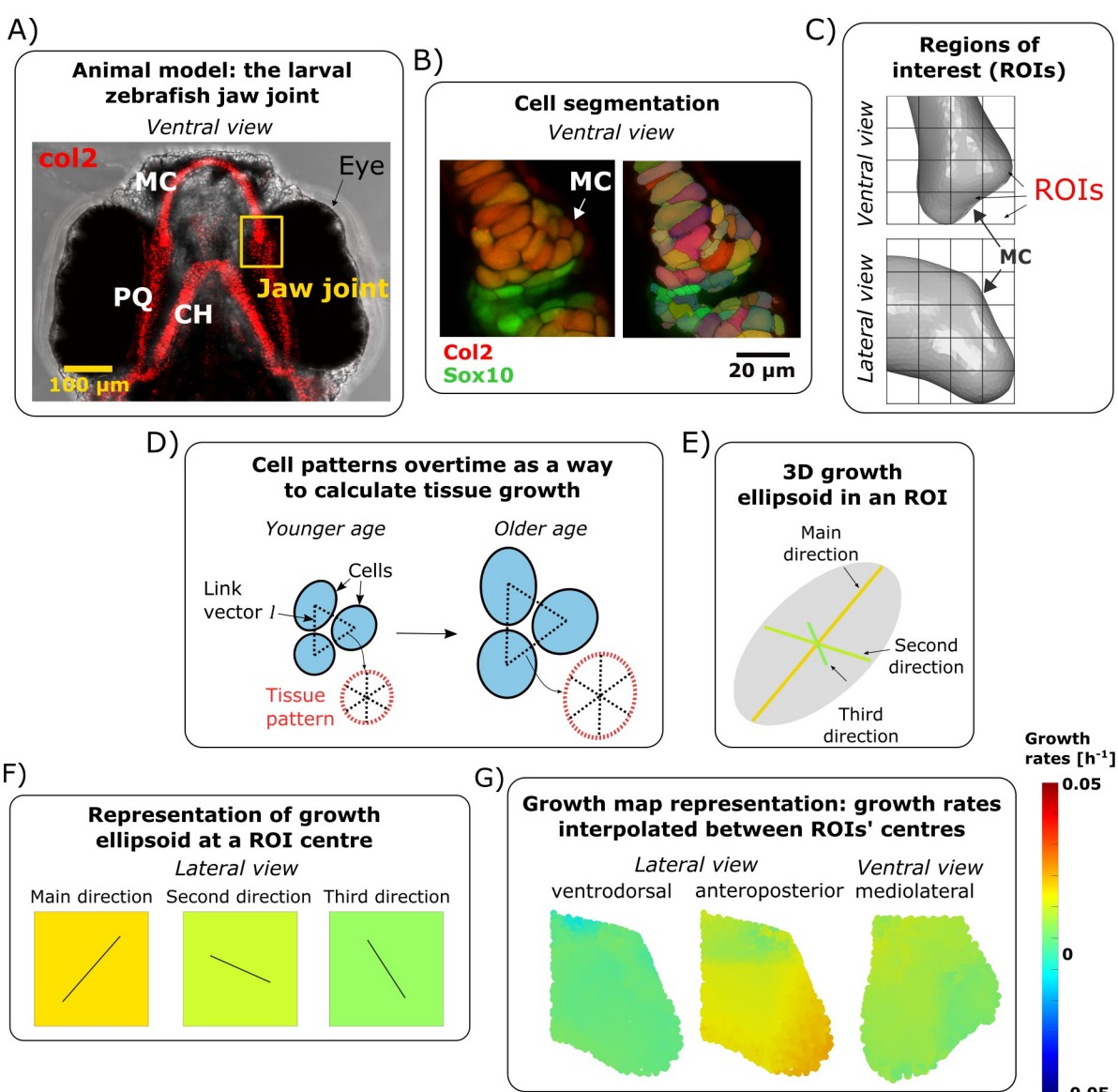

**Fig 5. Growth rate calculations.** (A) Brightfield ventral view of a 7 dpf zebrafish head expressing Tg(Col2a1aBAC:mCherry) cartilage marker showing the location of the jaw joint (yellow box). (B) Confocal ventral view of the jaw joint of a live 4.5 dpf zebrafish expressing the transgenic reporters Col2a1aBAC:mcherry (red) and -4.9sox10:eGFP (green) which mark cartilage chondrocytes. Cell segmentation result is shown on the right. (C) A grid marks out the regions (ROIs) of the anterior joint element in which growth is characterised. The length of each cube side is 15μm. (D) The position of cells with respect to each other forms a pattern. This pattern evolves over time and is used to characterise growth. (E) Computed growth is represented by an ellipsoid with orthogonal axes. The ellipsoid's radii correspond to the growth rates and their orientation to the direction of deformation. (F) At an ROI centre, growth rate is represented by the square's colour while the direction of growth is shown by solid black lines in the corresponding square. (G) Growth rates are interpolated between ROI centres to obtain the resulting growth maps. For simplicity, the orientation of deformation is approximated based on anatomical axes, actual orientations (as defined by the axes of growth ellipsoids) are provided in S4 Fig. MC: Meckel's cartilage, PQ: Palatoquadrate, CH: ceratohyal.

The position of cell centroids with respect to each other over time was used to calculate the local rate of deformation in cubic regions of interests (ROIs) shown in Fig 5C using the "statistical symmetrised velocity gradient" matrix developed by Graner et al. [63]. Graner et al. [63] defined the statistical symmetrised velocity gradient as a tool to describe at which rate and in which direction the pattern of the tissue deforms between consecutive timepoints $t$ and $t+\Delta t$.

**Table 1. Sample number per time window for growth rates analyses in the anterior joint element of free-to-move and immobilised larvae.**

|  | 4–4.5 dpf | 4.5–5 dpf | 5–5.5 dpf |
|---|---|---|---|
| Free-to-move | 7 | 7 | 7 |
| Immobilised | 8 | 6 | 7 |

The gradient quantifies local tissue distortions, such that if neighbouring cells grow or intercalate, or if extracellular matrix is built, the distance between cell centroids, and therefore the geometry of the tissue, change. The statistical symmetrised velocity gradient has units of $s^{-1}$ and is a statistical measurement of tissue changes equivalent to strain rate [63] as described in detail in S1 Text. Briefly, the relative position of cell centroids at a timepoint $t$ forms a pattern which can be described from the link vectors $l$ connecting neighbouring cell centroids (Fig 5D). Over time, the link vectors $l$ may change in length and direction, changing the pattern's overall geometry (Fig 5D). In our previous paper on free-to-move larvae at the same developmental stage, we showed that growth from 4–5 dpf could be accurately estimated using the velocity gradient methodology [47]. Our prior work also demonstrated that the direct inclusion of cell division was not essential for correct quantification of tissue growth rates and directions, due to the very low rate of proliferation over the time window assessed [47]. The cubic ROIs were of size length 15 μm and were mapped onto the MC joint element (Fig 5C). Calculations of the statistical symmetrised velocity gradient were computed for each ROI using the link vectors $l$ connecting the cells within the ROI to all their neighbours (whether or not these were part of the ROI).

After calculations, the local growth matrix (statistical symmetrised velocity gradient) can be represented by an ellipsoid whose axes (eigenvectors) represent the three directions for growth and their radius (eigenvalues) the rate of growth along these directions (Fig 5E). Growth maps displaying the local deformation rates and directions were generated from the local growth ellipsoids for each time window (Fig 5F and 5G). Interpolation between ROI centres was performed in Abaqus CAE (Dassault Systemes, 2019) by importing the growth maps as analytical mapped fields, as described in the "Simulating zebrafish jaw joint morphogenesis" section. For simplicity, ventrodorsal (VD)/anteroposterior (AP)/mediolateral (ML) growth is defined as the growth rates of the growth ellipsoid axis whose angle from the anatomical VD/AP/ML axis is the smallest amongst the three ellipsoid axes. For example, the main direction of deformation (major axis of the ellipsoid) tended to align most strongly with the VD axis, and therefore growth rates along main direction of deformation were called VD growth rates. All growth maps were analysed and reported following this terminology. The growth ellipsoid axes are displayed in S4 Fig, along with the angles between the growth ellipsoid axes and the anatomical VD/AP/ML axes. The number of samples analysed per time window are listed in Table 1. The p-value for growth rates mean differences along each direction for growth between free-to-move and immobilised groups for each time window were obtained by running Shapiro-wilk test of normality followed by Mann-Whitney U-test with Bonferroni adjustments for multiple comparisons.

## Average shape generation

Average shapes were generated at 4, 4.5 and 5 dpf for free-to-move and immobilised larvae following the methodology previously described [47] and described in brief below. Confocal image stacks of four to five larval zebrafish jaws (encapsulating the Meckel's cartilage, the palatoquadrate and the ceratohyal) from the transgenic line *Tg(col2a1aBAC:mCherry)* were taken

with a Leica SP8 confocal microscope at each time point in the ventral plane. A 3D Gaussian grey filter was applied to the image stacks in Fiji. Image stacks were imported into Mimics (Materialise NV, Leuven, Belgium) to be segmented. Only half-jaws (separated at the level of the midsagittal plane) were segmented. The segmented half-jaws were aligned in 3-matic (Materialise NV, Leuven, Belgium) using the joint line and retroarticular process (shape feature at the ventral side of the MC [62]). In MATLAB, segmented and aligned half-jaws were divided into slices in the transversal plane. For each slice, an average shape outline was generated in MATLAB from the shape vertices of each segmented half-jaw. Averaged shape outlines were saved as image stacks and imported into Mimics where the resultant average half-jaw shape was generated. The outlines of the average MC joint element were consistently cropped along the anteroposterior axis based on measured increases over time of the distances between the tracked cells and the joint line.

### Simulating zebrafish jaw joint morphogenesis

To validate that the growth rates computed from tracked cell activities drive the observed shape changes, growth was simulated in free-to-move and immobilised larvae using finite element (FE) methods implemented in Abaqus CAE. Growth in free-to-move and immobilised zebrafish jaws was simulated for each 12-hour interval time window (4–4.5 and 4.5–5 dpf) following the methodology previously published and described hereafter [47].

### Morphology generation

The morphologies as non-manifold assemblies were generated in Mimics at 4 and 4.5 dpf from the average half-jaw at the appropriate time point and the interzone which separates the Meckel's cartilage and palatoquadrate joint elements. The interzone was added as a volume filling the gap between the two joint elements using Boolean operations, with the interzone's external boundaries approximated based on imaging data [64]. Morphologies were meshed in 3-matic with ten node tetrahedral elements of approximate size 2 μm In Abaqus CAE, a finite element (FE) model for the starting point of each twelve-hour time window was created from the meshed assemblies.

### Material properties and boundary conditions

Cartilaginous regions were assigned homogeneous isotropic elastic material properties with Poisson's ratio of 0.3 and Young's Modulus (YM) as described in Table 2. The YM values were calculated based on nanoindentation measurements provided in S2 Text. When FE simulations of joint motion were performed using viscoelastic material properties or with linear elastic material properties, the pressure distributions were almost identical between the two types of model (S3 Text), giving us confidence that using linear elastic properties was appropriate for our simulations. No differences in YM were observed between hypertrophic and immature cartilage material properties (Fig B in S2 Text), and therefore all cartilaginous elements were assigned the same isotropic material properties with no distinction between regions. The

**Table 2. Zebrafish jaw cartilage Young's Moduli [kPa] in free-to-move and immobilised larvae used in FE models based on nanoindentation measurements.**

|              | 4–4.5 dpf | 4.5–5 dpf | 5–5.5 dpf |
| ------------ | --------- | --------- | --------- |
| Free-to-move | 142.01    | 142.01    | 142.01    |
| Immobilised  | 82.91     | 117.44    | 151.96    |

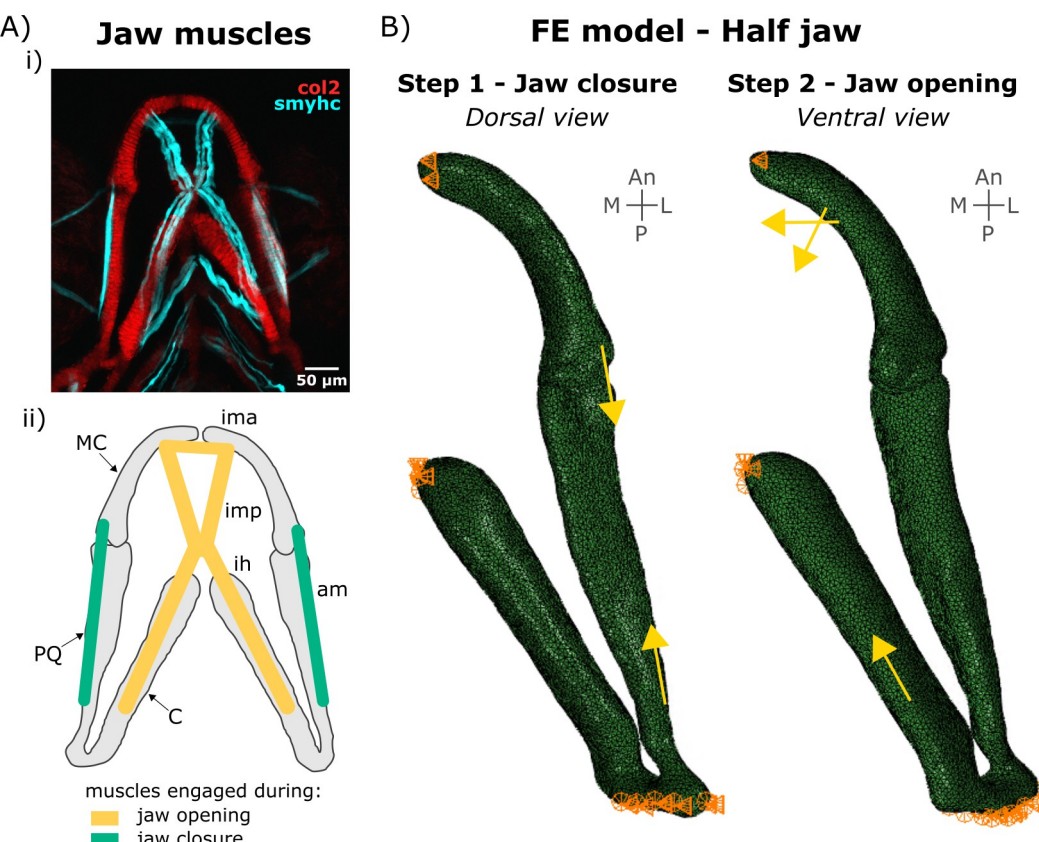

**Fig 6. Boundary and loading conditions in the FE model.** A) Lower jaw muscles. (i) Maximum projection of ventral confocal image stacks expressing Col2a1aBAC:mcherry (red) and smyhc1:EGFP (cyan) of a 4 dpf larva. (ii) Schematic of the muscles engaged during lower jaw opening (yellow) and closure (green) in the ventral plane. am: adductor mandibularis, ih: interhyal, ima: intermandibularis anterior, imp: intermandibularis posterior. B) Half jaw finite element (FE) model of jaw closure and opening with boundary conditions and muscle loads. An: Anterior, CH: ceratohyal, L: Lateral, M: Medial, MC: Meckel's cartilage, P: Posterior. PQ: palatoquadrate.

interzone was assigned isotropic elastic material properties with Poisson's ratio 0.3 and YM set at 0.025% of the cartilaginous YM [65].

Physiological boundary conditions were applied as previously published [47] and as shown in Fig 6B. The anterior end of the ceratohyal was fixed in all directions (Fig 6A) and translations in the lateromedial direction of the anterior ends of the Meckel's cartilage were prevented to maintain the symmetry with the missing half-jaw (Fig 6B). Only anteroposterior translations of the posterior end of the palatoquadrate, where the lower jaw connects to the rest of the craniofacial skeleton, were allowed (Fig 6B).

### Integration of growth maps into FEA

For each 12-hour period, growth strains derived from the growth maps were imported into Abaqus CAE as three distinct analytical mapped fields (one for each axis of the growth ellipsoids) and applied to the model. The coordinates of the ROI centres were assigned the calculated strains (strain = growth rates * time interval) and interpolation was performed between ROI centres to assign strains to each element. Local material orientations matching the local directions for growth were assigned to the joint elements. Elements whose nodes' coordinates were contained within an ROI were all assigned the directions for growth of this ROI.

The Abaqus user subroutine UEXPAN was used to apply spatially varying expansion based on the strain fields to provide a prediction of growth and shape for each time-window. Outlines of the simulated shapes were obtained for both free-to-move and immobilised larvae for the anterior MC joint element as shown in full in Fig B in S1. The global patterns of free-to-move and immobilised jaw joint morphogenesis were correctly simulated using the growth rates obtained from cell-level data, including the observed depth increases in free-to-move controls but not in immobilised, and higher length increases in free-to-move controls than in immobilised, as shown in S1B Fig.

## Simulating zebrafish jaw movements

Jaw movement simulations were performed on free-to-move 4 and 4.5 dpf FE models in Abaqus CAE. Muscle contractions engaged during opening/closure shown in Fig 6A were applied to the models as point loads in fixed directions under quasi-static conditions following a previously published model [64] (Fig 6B). Muscle attachment points and directions were estimated from confocal scans of double transgenic *Tg(col2a1aBAC:mCherry;smyhc1:EGFP)* larvae (Fig 6Ai). Muscle forces enabling physiological jaw displacement (jaw opening of 37.2 μm based on the average jaw displacement for 5 dpf larvae [19]) were used and are listed in Table 3. The ratio of muscle forces was obtained from [19] based on the number of fibres of each muscle engaged during jaw motion measured from confocal scans of double transgenic *Tg(col2a1aBAC:mCherry;smyhc1:EGFP)* larvae. Jaw closure and opening were simulated in subsequent steps with each step decomposed into five increments. Hydrostatic stress fields were extracted into MATLAB for each time increment from peak closure to peak opening.

## Investigating zebrafish jaw joint mechanoregulation

Hydrostatic stress fields were extracted from jaw movement simulations at 4 and 4.5 dpf and integrated into 4–4.5 and 4.5–5 dpf growth simulations respectively. The growth rates from the immobilised larvae were used as the baseline biological contribution to growth, and were applied to the free-to-move shapes (called $G_{imm}$) (Fig 7A). Hydrostatic stress fields (Fig 7B) were used to supplement $G_{imm}$ providing new mechanobiological growth maps which were applied in growth simulations (Fig 7C). Two different methods to calculate the mechanobiological growth maps were implemented to test different hypotheses.

Firstly, we tested the hypothesis that average hydrostatic stresses over the loading history direct jaw joint morphogenesis with compressive stresses promoting growth. The average hydrostatic stress field across loading history $S$ was calculated from the hydrostatic stress fields of all step increments in MATLAB (Fig 7A method 1). At each node of the model mesh the average hydrostatic stress is:

$$S = \frac{\sum_{i=1}^{N} \boldsymbol{\sigma}_{hi}}{N}$$

where $\boldsymbol{\sigma}_h$ is the hydrostatic stress field at the considered node and $N$ the number of step

**Table 3. Muscle forces [nN] in the lower jaw enabling physiological movement in FE simulations.**

|                                   | 4 dpf | 4.5 dpf |
|-----------------------------------|-------|---------|
| adductor mandibularis (am)        | 2.84  | 4.35    |
| intermandibularis anterior (ima)  | 1.25  | 2.90    |
| intermandibularis posterior (imp) | 1.50  | 3.47    |
| interhyal (ih)                    | 1.50  | 3.47    |

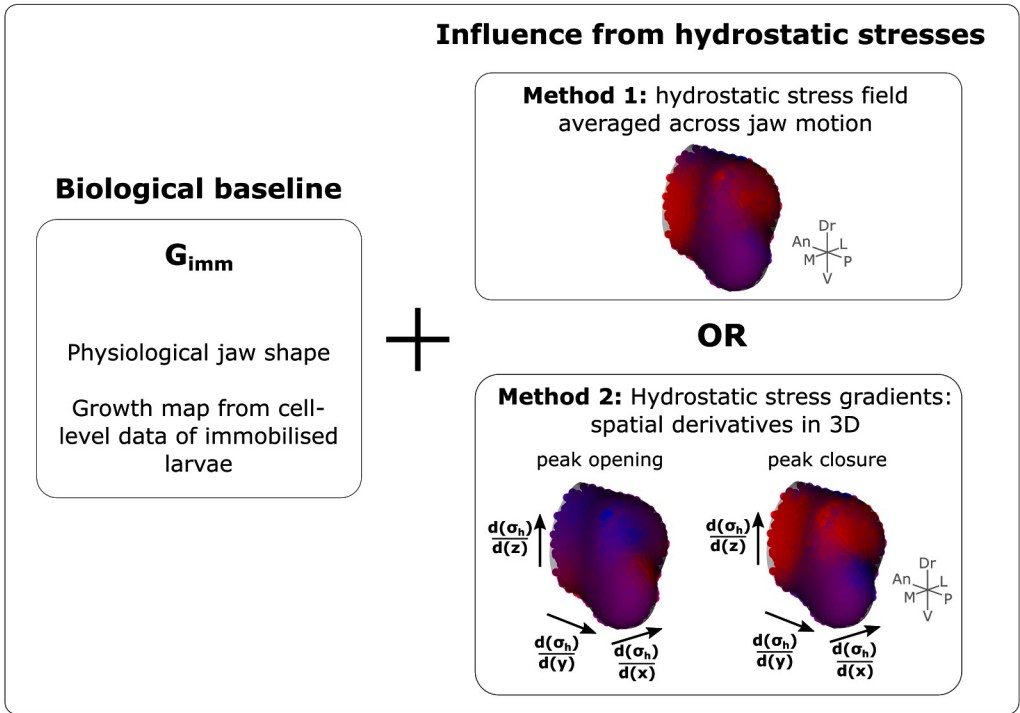

**A) Growth maps from cell-level data**

**B) Jaw movement simulations**

**C) Mechanobiological growth maps**

**Fig 7. Process of integration of mechanical stimuli arising from movements into mechanobiological growth maps of zebrafish jaw joint morphogenesis.** A) Terminology of two growth maps obtained from cell-level data: $G_{free}$ and $G_{imm}$. B) Jaw opening and closure were simulated, and the hydrostatic stress fields were extracted. C) The biological baseline $G_{imm}$ was influenced by stresses arising from jaw motion either using the stress field averaged across jaw motion (method 1), or the hydrostatic stress gradients at peak jaw opening and peak closure (method 2).

increments from peak opening to peak closure. To test the hypothesis that compression promotes growth [66], a mechanobiological growth map $\mathbf{G_{ave}}$ was calculated. At each node we set:

$$\mathbf{G_{ave}} = \mathbf{G_{imm}} - a.\mathbf{S}.\mathbf{I_3}$$

where $a$ is a mechanoregulatory growth modulating variable which influences the impact of the average hydrostatic stress field $\mathbf{S}$ on the mechanobiological growth map, and was considered to account for the amount of loading over twelve hours (approximatively 57 thousand jaw openings between 4 and 4.5 dpf) and $\mathbf{I_3}$ is the identity matrix of order 3x3. A sensitivity analysis was performed to determine the most appropriate value for $a$. Using the 4–4.5 dpf simulation, $a$ was incrementally increased from 1e9 $m^2N^{-1}s^{-1}$ to 4e9 $m^2N^{-1}s^{-1}$. At the lowest value, there was minimal mechanobiological growth, while the maximum value led to obvious joint overgrowth, as shown in S5 Fig. The modulating variable value which best predicted physiological MC depth growth was chosen ($a$ = 3e9 $m^2N^{-1}s^{-1}$). The same value for $a$ was used for the 4.5–5 dpf simulation.

The contribution of the average hydrostatic stress field to growth was isotropic, with the same hydrostatic stress field and modulating variable applied along all directions. The obtained mechanobiological growth maps $\mathbf{G_{ave}}$ was qualitatively compared to the growth map calculated from free-to-move larvae $\mathbf{G_{free}}$. Morphogenesis simulations were run in Abaqus CAE using $\mathbf{G_{ave}}$ based on the same methodology explained in section "Simulating zebrafish jaw joint morphogenesis" and qualitatively compared to growth simulations using $\mathbf{G_{free}}$.

Next, we tested the hypothesis that the difference between stress levels at each side of the rudiment compared to its opposite one coupled with the change in this difference over the opening and closing cycle of the joint promotes jaw joint anisotropy. The rationale for this hypothesis was that along the ventrodorsal axis, where growth of the joint shape was the most pronounced in normally moving fish, a gradient in stress from tension to compression was observed during jaw opening which was reversed during jaw closure. For each node of the mesh model, the hydrostatic stress gradients $\nabla\boldsymbol{\sigma_h}$ along each anatomical axes $\mathbf{x}$, $\mathbf{y}$ and $\mathbf{z}$ were calculated at peak closure and peak opening (the most extreme scenario) in MATLAB [67] (Fig 7C method 2). Here, these hydrostatic stress gradients were expressed in the form of diagonal matrices:

$$\nabla\boldsymbol{\sigma_h} = \begin{matrix} \dfrac{\partial\sigma_h}{\partial\boldsymbol{x}} & 0 & 0 \\[2mm] 0 & \dfrac{\partial\sigma_h}{\partial\boldsymbol{y}} & 0 \\[2mm] 0 & 0 & \dfrac{\partial\sigma_h}{\partial\boldsymbol{z}} \end{matrix}$$

where $\sigma_h$ is the hydrostatic stress at the considered node and (x, y, z) are the anatomical axes.

To account for the temporal changes in stress gradients over jaw motion with the hypothesis that a switch in the direction of the gradients promotes growth, changes to the hydrostatic stress gradients between peak opening to peak closure were calculated: the absolute value of the difference between peak opening and peak closure was taken. A new growth map $\mathbf{G_{grad}}$ was calculated at each node of the mesh model based on the following equation:

$$\mathbf{G_{grad}} = \mathbf{G_{imm}} + b.|\nabla\nabla\boldsymbol{\sigma}_{\mathbf{h\ peak\ opening}} - \nabla\nabla\boldsymbol{\sigma}_{\mathbf{h\ peak\ closure}}|$$

where $\mathbf{G_{imm}}$ is the growth tensor obtained from immobilised larvae at the considered node, $b$ is a mechanoregulatory growth modulating variable which influences the impact of the hydrostatic stress gradients on the mechanobiological growth map (units [$m^3N^{-1}s^{-1}$]) and $\nabla\nabla\boldsymbol{\sigma}_{\mathbf{h\ peak}}$

**Table 4. Overview of terminologies and methods used for different growth maps.**

| Growth map | Description | Contribution of mechanical loads | Hypothesis tested | Mechanoregulatory growth modulating variable |
|---|---|---|---|---|
| $G_{free}$ | Growth rates obtained from free-to-move cell-level data | Normal baseline | Growth simulations with tracked free to move cell data accurately predict free to move morphogenesis | - |
| $G_{imm}$ | Growth rates obtained from immobilised cell-level data | None | Growth simulations with tracked immobilised cell data do not accurately predict free to move morphogenesis | - |
| $G_{ave}$ | Mechanobiological growth maps | Average hydrostatic field across load history | Compression levels promote growth | $a = 3e9 \text{ m}^2.\text{N}^{-1}\text{s}^{-1}$ |
| $G_{grad}$ | Mechanobiological growth maps | Hydrostatic stress gradients | Hydrostatic stress gradients promote growth anisotropy | $b = 5e10 \text{ m}^3.\text{N}^{-1}.\text{s}^{-1}$ |
| $G_{comp\_grad}$ | | Compressive hydrostatic stress gradients | Compressive hydrostatic stress gradients promote growth anisotropy | $b = 5e10 \text{ m}^3.\text{N}^{-1}.\text{s}^{-1}$ |

$_{\text{opening}}$ and $\nabla\nabla\sigma_{\text{h peak closure}}$ are the hydrostatic stress gradients at peak opening and peak closure respectively. A sensitivity analysis was again performed to find the most appropriate value for $b$. For the 4–4.5 dpf simulation, the value of $b$ was incrementally increased from $1e10 \text{ m}^3\text{N}^{-1}\text{s}^{-1}$ to $1.5e11 \text{ m}^3\text{N}^{-1}\text{s}^{-1}$, leading to the effects on mechanobiologically controlled growth shown in S5 Fig. The modulating variable value which best predicted physiological MC depth growth was chosen ($b = 5e10 \text{ m}^3\text{N}^{-1}\text{s}^{-1}$). The same value of $b$ was used for the 4.5–5 dpf simulation. For the gradient simulations, the contribution of mechanical fields to growth was anisotropic: the same modulating variable value $b$ was used along all anatomical axes, but the stress gradients varied between axes. The newly obtained mechanobiological growth map $G_{grad}$ was qualitatively compared to the growth map calculated from free-to-move larvae $G_{free}$. Morphogenesis simulations were run in Abaqus CAE using $G_{grad}$ and qualitatively compared to growth simulations using $G_{free}$ (Fig 7F). To asses the contribution of only compressive hydrostatic stress gradients to joint morphogenesis, we calculated a new mechanobiological growth map $G_{comp\_grad}$ using the same methodology than for $G_{grad}$ except that only the compressive hydrostatic stresses were considered. The same value for the mechanoregulatory growth modulating variable $b$ was used ($b = 5e10 \text{ m}^3\text{N}^{-1}\text{s}^{-1}$). Terminologies and methods for calculation of the mechanobiological growth maps are summarised in Table 4.

## Supporting information

**S1 Fig. Growth simulations from cell-level data.**
(DOCX)

**S2 Fig. Mechanobiological simulations of zebrafish larval jaw joint morphogenesis from 4–4.5 dpf incorporating different biological and mechanobiological contributions.**
(DOCX)

**S3 Fig. Ventral views of the hydrostatic stress fields averaged across load history and at peak opening and peak closure.**
(DOCX)

**S4 Fig. Jaw joint growth orientations.**
(DOCX)

**S5 Fig. Sensitivity analyses of mechanoregulatory growth modulating variables affecting growth rates from 4 to 4.5 dpf.**
(DOCX)

**S1 Text. Growth rates calculations from cell positional information.**
(DOCX)

**S2 Text. Material properties characterisation using nano-indentation.**
(DOCX)

**S3 Text. Comparison between linear elastic and viscoelastic material properties in jaw movement simulations.**
(DOCX)

## Acknowledgments

We thank Dr James Monsen for providing the methodology and MATLAB script which was used for generating average shapes. We would like to thank Mr Mat Green for zebrafish husbandry and the staff of the Wolfson Bioimaging centre Bristol for imaging support. We thank Dr David Labonte and his team for sharing nano-indentation equipment with us and Dr Maria Kaimaki for her valuable help during nano-indentation experiments.

## Author Contributions

**Conceptualization:** Josepha Godivier, Elizabeth A. Lawrence, Chrissy L. Hammond, Niamh C. Nowlan.

**Data curation:** Elizabeth A. Lawrence, Mengdi Wang.

**Formal analysis:** Josepha Godivier.

**Funding acquisition:** Chrissy L. Hammond, Niamh C. Nowlan.

**Investigation:** Josepha Godivier.

**Methodology:** Josepha Godivier, Elizabeth A. Lawrence, Chrissy L. Hammond, Niamh C. Nowlan.

**Project administration:** Niamh C. Nowlan.

**Software:** Josepha Godivier.

**Supervision:** Chrissy L. Hammond, Niamh C. Nowlan.

**Validation:** Josepha Godivier.

**Visualization:** Josepha Godivier.

**Writing – original draft:** Josepha Godivier.

**Writing – review & editing:** Josepha Godivier, Elizabeth A. Lawrence, Mengdi Wang, Chrissy L. Hammond, Niamh C. Nowlan.

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
