## [Decision Letter · Decision Letter 0]

9 May 2023

Dear Dr Nowlan,

Thank you very much for submitting your manuscript "Cyclical compression loading is the dominant mechanoregulator of synovial joint morphogenesis" for consideration at PLOS Computational Biology.

As with all papers reviewed by the journal, your manuscript was reviewed by members of the editorial board and by several independent reviewers. In light of the reviews (below this email), we would like to invite the resubmission of a significantly-revised version that takes into account the reviewers' comments.

I would like to particularly draw attention to the data and code availability as a reviewer has indicated that sufficient details have not been provided for reproducing the results.

We cannot make any decision about publication until we have seen the revised manuscript and your response to the reviewers' comments. Your revised manuscript is also likely to be sent to reviewers for further evaluation.

Sincerely,

Kiran Raosaheb Patil, Ph.D.

Section Editor

PLOS Computational Biology

I would like to particularly draw attention to the data and code availability as a reviewer has indicated that sufficient details have not been provided for reproducing the results.

Reviewer's Responses to Questions

**Comments to the Authors:**

Reviewer #1: Cyclical compression loading is the dominant mechanoregulator of synovial joint morphogenesis

This paper examines the role of mechanics in early zebrafish jaw joint growth, combining experiments with computational modeling.

General:

It seems that the basic question is: can you predict dorsal-ventral growth anisotropy with mechanical stimulus? You found that dorsal-ventral growth was greater than other directions and then searched for a mechanical stimulus that would give you this. Is this correct? If so, the paper can be made much more clear throughout.

Title: The title is too overblown. Morphogenesis (joint shape) was not really investigated in this paper. What about “Role of cyclic compression in anisotropic growth in the larval zebrafish jaw”? This brings it back to what was actually studied.

This is a lot of detailed work that went into this paper. While it is not necessary to include all the details, you need to include sufficient that we can understand what is going on. For example, what are the loading conditions on your FEA model? Over-complication seems to be a trait of this paper. Sometimes by not describing well, you overcomplicate things. It is currently a very tough paper to grasp. It take a few thorough reads to even understand what you are doing. Let’s try to make it more accessible. De-complication is a theme of many of the edits suggested below. You want the reader to skim it once and say “wow! That is really interesting that they found mechanics may drive anisotropic growth. What a clever way to figure this out.” Then they read it again for the details of exactly what you did. Right now one gets stuck in the details, but many of the details are not that relevant and relevant details are left out. Specific comments are below. Try to de-complicate and clarify the writing.

Specific:

Line 21: Change to read “we integrated cell growth data from experiments into a finite element model of zebrafish jaw growth to identify links between mechanical stimuli and anisotropic growth.”

Line 32: You imply that “magnitude of compression” is a statically applied load. It is not, it could be cyclic, but you just are capturing the magnitude.

Line 34 “However, the dynamic changes caused by cyclic compression” ….isn’t cyclic compression always applied? It is just what you are choosing to look at. All the loading is dynamic.

Line 35: “sizes and shapes of joints were correctly simulated” Unclear that shape was robustly analyzed and size is fairly arbitrary and can be set by a multiplicative constant. Perhaps just stick with “anisotropic growth of the cartilage was correctly simulated”.

Line 36: Change the sentence “We conclude….” To “We conclude that the cyclic motion of the jaw joint contributes to anisotropic growth of the cartilage.” (or rudiement if it isn’t cartilage yet).

Line 38: It is not a “fundamental advance”. Change this sentence to read “This study demonstrates the importance of mechanical stimuli in regulating anisotrpic growth in early development.”

Line 118: One should be very careful linking the stimuli for growth and ossification with the stimlui for cartilage growth. Make it clear that “octahedral shear stress” is for hypertrophy and ossification of the cartilage. Yes, this is growth, but different than what you are looking at in this paper where you hardly have any matrix at the timepoint of interest…we are far away from hypertrophy and ossification.

Line 123: It is not super convincing that cell-level dynamics contribute critically to the study. Cell-level dynamics provides the mechanism through with dorsal-ventral anisotropic growth occurs, but you are not directly modeling cell migration, cell proliferation, etc. Be more precise about what you mean “cell-level dynamics” and how it is “necessary”.

Line 144: Unclear what “magnitude of compression” is from “cyclic compression”. Cyclic compression is always applied, you are just looking for a stimulus that matches anisotropic growth. You imply “magnitude of compression” is static and “cyclic compression” is the answer. This does not match what you did: compression is always cyclic in the model, correct? Don’t we already know that cyclic compression is critical for cartilage growth? Clarify what is novel.

Line 152: “no significant differences in shape” Are you really studying morphogenesis? Or just anisotropic growth?

Results: The authors have beautiful 3D data that they shrink down to simplified 2D line drawings. They are encouraged to think about how they might better represent the 3D growth.

Figure 1D: The results showing length, depth, and width demonstrates the anisotropy. Do you need cell-level dynamics data to determine that from 4-5dpf, it grows a little in length, a lot in depth, and not at all in width. Can’t a ruler do this?

Figure 2: A) is very helpful for orientation. B) does not add much to what we already know from Figure 1. The model does what you tell it to. Is this necessary to show? Perhaps move this figure to Supplementary. Unclear why the outlined shape does not change for mediolateral growth. How can you see it in the dorsal-ventral/anterior-posterior plane? Why not show a plane that actually has the mediolateral direction in it? The shape of your outline will change.

E) Unclear what the violin plots are showing. How many data points do you have here? Are these the growth rates of ROIs? Again, doesn’t Figure 1 show the same information? Is this an average or a representative sample? It is a shame that the authors have to shrink beautiful 3D data onto a 2D plane. Are there better ways to represent the 3D data?

Paragraph starting in 175. This says we measured anisotropic growth and modeled it to ensure it gave us what we wanted (i.e. we could grow a model by the growth rates calculated and obtain the “right result”). This is all fairly arbitrary, isn’t it? Because you are not looking for changes in shape, in which patterns of high growth may be regional, but looking for anisotropic growth, why is magnitude so interesting? You can tune it to give you the magnitude you want. There are surely enough constant multipliers in your model to do this. I think the entire paragraph can be deleted with little lost. If it adds something more than Figure 1, make it clear what is new that we don’t know from Figure 1.

Line 206: Is Gimm the average of the immobilized growth rates?

Line 210: Can we rename the stimuli: 1) average hydrostatic stress; 2) peak hydrostatic stress gradient; 3) peak hydrostatic compressive stress gradient? If you don’t like these names, determine better, more concise names for the three stimuli. “Dynamic” and “time evolution” don’t see appropriate since loading was dynamic for all of them. If anything “average” captures more the dynamic than analyzing just at peak.

It is unclear why you would average hydrostatic stress if you know the cartilage is in bending; the average cancels out any dorsal/ventral differences. This stimulus can be removed from the paper. It is not clear why it was chosen. Other papers that have looked at “average stimulus” are not examining bending conditions, rather multiple compressive loads.

Figure 2 BCD and Figure 3A present the growth planes in different orders. Unnecessarily confusing. It is also not clear why these are not the same. Clarify. Figure 1C has free-to-move (control) on the left and immobilized on the right. Figure 2B has free-to-move on the top and immobilized on the bottom. Figure 2A has immobilized on the top and free to move on the bottom. This is all unnessarily confusing. Be consistent in the layout so we are clear what we are looking at. Shouldn’t the outlines in 3Aiii match the outlines in 1c? As in all figures, you cannot show ML growth in an AP/DV plane. Show an appropriate cross section. Rename stimuli in B, C, D. What you are trying to do is match the orange and the green in 3bii, correct? Are 3C and 3D different? They look very similar. Rename G_dynm as all loads are dynamics (as opposed to static).

Line 268 It is unclear why you chose to compute stress gradients rather than directly computing the time difference between stress maps.

Line 290: Remove language referring to “dynamic shift”. This is not good terminology.

Line 366: perhaps not a fundamental advance. …just an understanding.

Line 312: It is not clear that you have definitively shown cyclic compression rather than cyclic stress.

Line 375: Anesthetized (immobilized larvae) are presumably not eating. Could lack of nutrients explain lack of growth?

Line 380: The method for determining cell growth is not clear without reading the authors’ previous J Anatomy paper. Line 388: bout how many cells are being analyzed at this timepoint? From Figure1 in the J Anatomy paper, it seems like 15-20 cells?

Line 391: “The position of cell centroids with respect to each other over time was used to calculated the local rate of deformation.” Is this a good measure of growth? What if cells proliferate, migrate without growing, grow in size but centroids stay in the same place, or intercalate? At the time point of interest, 4-5 days, is there any cell proliferation? Are you assuming all growth is due to growth in the size of the cells. Even though this is from the previous paper, it is not well justified in the previous paper either. Provide better assurance that the “statistical velocity gradient” is a good measure of local deformation at this time point.

From the J Anat paper, each cell is about 10 microns and the ROI is 15 microns. How does this work? You are tracking growth of individual cells and mapping this to an ROI? How many ROIs do you have in 3D? How many cells? In this paper, you established a method to identify cells, identify their centroid at each time point, match cells across timepoints. Multiple cells contribute to a statistical velocity gradient, but the “statistical velocity gradient” is not clearly defined (only referenced to another paper). The process of segmenting cells � obtaining growth directions for ROI is not clear. You need to include enough detail about the “cell dynamics” that we understand what you are doing without having to read multiple previous papers.

Line 395: It is unclear why the cubic ROIs of line 394 suddenly become ellipsoidal ROIs in line 395. Couldn’t you represent this growth as axes on a cube?

Figure S3. Why aren’t VD growth vectors in the VD direction? In day 4.5-5 they are, but in day 4-4.5 some of them seem more AP. How can you show ML growth in this plane? Why not show it in a plane that shows ML? One would expect all the points to be dots in this view (as seen in the 4.5-5 pdf), which isn’t a helpful vector. The scale bar in this picture is confusing. It seems like the entire region is about 60 microns long, which from Figure 1 in the J Anat paper is probably about 6 cells. There are way more than 6 cells represented, which makes it seem this is a projection through the entire stack, yet the caption says “results are displayed in one section in the mid-lateral plane”. If an angle is > 45 wouldn’t that indicate that it was being assigned to the wrong axis? It seems like you assigned the directions based on which anatomical axis it was closest to.

Line 411: How do you ensure alignment of the images? Cross sectional shape depends on the orientation of the object in the image. Line 417: image stacks were imported, but how were they aligned to be in the same direction? It seems a shame to represent your 3D images with 2D outlines. There are methods to register 3D shapes and measure differences. Justify the use of 2D outlines.

Line 424: This entire section of “measuring modulus” seems fairly irrelevant when you just use a linear elastic model. An 8 micron radius indentor is the same size as the cells. With only 1 micron depth, the contact area is much smaller than the cell. This implies you are just measuring stiffness of a single cell for each of the larvae? Is this really helpful when you end up using a linear elastic modulus? I’d suggest you move this to Supplementary Information, if you want to keep it at all. (I’m not sure measuring stiffness of a single cell is really that relevant to what you are doing.)

Line 439: What does “all sections containing relevant cartilage regions” mean? What is a section? What is a relevant cartilage region? It seems like you pressed on one cell in each larva.

Line 458: In this section you put growth data from your experiments in your model and it gives back growth that you tell it. This seems like a check to make sure your growth calculations were correct. There are no results from this. You need this section for 2 reasons: 1) use this section to explain clearly how you calculate Gimm and what it means. 2) how you calculate Gfree and that you will use it as a comparison for the simulations (i.e. to “check” your answer).

Line 482: The loading conditions on the finite element model are not clear. Figure 6 does not indicate the loading conditions applied. Were forces or displacements applied? What was the direction of force? Where was it fixed? How fast was the load applied? Was it applied as a series of quasi static loads or as a dynamic load? Instead of Figure 6, it would be helpful to have the finite element model with the boundary conditions and loading conditions indicated.

Table 3: How was ratio of muscle forces determined?

Line 489: Why do you need stress and strain fields? It is a linear elastic model with homogenous material properties. They’ll tell you the same thing. (as you report in line 506). You don’t need S6. Just use hydrostatic stress forget the strain.

Line 500: What are “loading fields”?

Line 508: Because the “loading history” is not clear (see above comment), it is not clear how the mechanobiological growth is calculated. How many step increments (N) were applied? This seems to indicate jaw opening and closing is applied as quasi-static. More information on the loading conditions are required. Because you know that bending occurs, it seems rather silly to look at an average hydrostatic stress. The tension and compression experienced throughout opening and closing will cancel each other out. Average does not represent anything (as you found out). If you are only concerned about compression, why would you average it and then only look at compression. It is suggested that solid rationale for this choice of stimulus be made or remove it.

Line 516: a is a constant that balances out the influence of mechanics relative to biological growth. Is that correct? You chose it such that the amount of growth matched your experiment?

Figure 7 does not clarify anything. It is suggested that you have one figure with the loads/boundary conditions clearly indicated on the FE model. You delete Method 1. You clarify what a hydrostatic stress gradient is. D) “time evolution of hydrostatic stress gradient” is unclear

E) This should be one figure showing the hydrostatic stress stimulus in the various planes, clearly labeled. F) not clear what this is showing.

Line 545: It is unclear how a stress gradient captures the “dynamic changes in hydrostatic stress”. Provide more explicit details on how the gradient was calculated. Are the results different if you just evaluate the maximum hydrostatic compressive stress during a complete load cycle? Other work has used max hydrostatic compressive stress as a mechanobiological stimulus for decades. It is not clear why this is called “dynamic”. It is just at the peaks so just picking up when the gradient is largest. You chose the gradient because you needed a stimulus that varied in the D/V direction. What does hydrostatic stress gradient mean physiologically? What would this translate to at the cell

Line 549: Is this the absolute value of the difference of the gradients? Things are getting confusing. Is it really a time evolution if you just look at peak opening and closing?

Line 555: Check units of the constant. Do you need sec in there?

Line 565: This is unclear. Did you calculate the absolute value of the gradients of only the compressive stresses? Is Figure 7, Method 2 showing the stress gradients in the x,y directions? It would help if you provide a figure (can be in Supplemental) where you show the hydrostatic stress in the two planes of interest (like Figure 4B, but showing the M/L plane as well), then the hydrostatic stress gradients in the 3 directions (2 can be shown in one plane and M/L in another plane), then the hydrostatic compressive stress gradient in the 3 directions. This would really help show what is happening with the manipulations. This likely means adding to figure 4. 4A can be discarded unless you have a good rationale for trying this.4B is likely max tension/compressive stress. Add to this the gradients in the different planes so we can see the stimulus as contour plots.

Table 4: Remove “time evolution”. This is not an accurate descriptor.

Reviewer #2: This work presents an interesting multidisciplinar work, where experiments and computational simulations are combined in order to identify links between the

mechanical stimuli arising from movement and patterns of growth.

Although the work is really very well presented and very interesting, there are some relevant aspects that require further explanation or justification:

1) Authors indicate in the paper that they use an integrated cell-level data into a novel

mechanobiological model. It is not clear for me what they mean "an integrated cell-level data".

2) Authors conclude that the application of cyclic compression, rather than the cyclical switch from compression to tension, is likely to be the key contributor to jaw joint morphogenesis. Why do you think it is happening? Could it be due to the fluid flow associated to the compression loads? I think authors should discuss further why they think which is the local mechanobiological effect that could be more relevant for joint morphogenesis.

3) From the FE-based simulations, it is not clear for me which kind of elements and materials have been used for the simulations. In my opinion, the FE model should be described providing much more data, such as, the type of element used, the constitutive law used, boundary conditions and interaction constraints.

Reviewer #3: The goal of this work was to directly identify the relationship between biomechanical stimuli due to embryonic movement and joint morphogenesis using zebrafish jaw as a model system to directly measure cell activity and mechanical environment, and implemented these measures in a mechanobiological model of jaw morphogenesis. This paper offers a very strong background and motivation for quantification of biological stimuli and spatial analyses to augment understanding morphogenesis of joints and biomechanical predictive models of this process. The methods are appropriate and overall this is a very strong contribution to the literature.

The comparison between the dynamic simulations with and without inclusion of the tensile contribution is interesting. Because the simulations that do not consider tension capture the morphological development better, the authors conclude that the tension is not a large contributor. However, the tension appears to be a real condition experienced by the developing jaw, even though including it in the model leads to worse predictions – this could equally be a result of some error in the model itself in terms of how tensile stimuli are represented. Could the authors expand on this point in the discussion?

**Have the authors made all data and (if applicable) computational code underlying the findings in their manuscript fully available?**

Reviewer #1: Yes

Reviewer #2: **No: **I think further information should be given, so other authors could replicate the model.

Reviewer #3: Yes

PLOS authors have the option to publish the peer review history of their article (what does this mean?). If published, this will include your full peer review and any attached files.

Reviewer #1: No

Reviewer #2: No

Reviewer #3: No
---

## [Decision Letter · Decision Letter 1]

18 Jan 2024

Dear Dr Nowlan,

We are pleased to inform you that your manuscript 'Compressive stress gradients direct mechanoregulation of anisotropic growth in the zebrafish jaw joint' has been provisionally accepted for publication in PLOS Computational Biology.

Best regards,

Jason M. Haugh

Section Editor

PLOS Computational Biology

Kiran Patil

Section Editor

PLOS Computational Biology

Reviewer's Responses to Questions

**Comments to the Authors:**

Reviewer #1: The authors have satisfied all recommendations and the paper is much improved. Congratulations.

Reviewer #2: I think authors have improved the presentation of the work.

Reviewer #3: The authors have completed a robust revision, and have adequately addressed this reviewers concerns.

**Have the authors made all data and (if applicable) computational code underlying the findings in their manuscript fully available?**

Reviewer #1: Yes

Reviewer #2: Yes

Reviewer #3: None

PLOS authors have the option to publish the peer review history of their article (what does this mean?). If published, this will include your full peer review and any attached files.

Reviewer #1: No

Reviewer #2: No

Reviewer #3: No

---

## [Editor Report · Acceptance letter]

4 Feb 2024

PCOMPBIOL-D-23-00217R1 

Compressive stress gradients direct mechanoregulation of anisotropic growth in the zebrafish jaw joint

Dear Dr Nowlan,

I am pleased to inform you that your manuscript has been formally accepted for publication in PLOS Computational Biology. Your manuscript is now with our production department and you will be notified of the publication date in due course.

With kind regards,

Zsofia Freund
